# Difference in Intraspecies Transmissibility of Severe Fever with Thrombocytopenia Syndrome Virus Depending on Abrogating Type 1 Interferon Signaling in Mice

**DOI:** 10.3390/v16030401

**Published:** 2024-03-05

**Authors:** Byungkwan Oh, Seok-Chan Park, Myeon-Sik Yang, Daram Yang, Gaeul Ham, Dongseob Tark, Myung Jo You, Sang-Ik Oh, Bumseok Kim

**Affiliations:** 1Biosafety Research Institute, College of Veterinary Medicine, Jeonbuk National University, Iksan 54596, Republic of Korea; guroom2@gmail.com (B.O.); vetpath@jbnu.ac.kr (S.-C.P.); 111@kongju.ac.kr (M.-S.Y.); ddddaram@naver.com (D.Y.); ge990915@naver.com (G.H.); tick@jbnu.ac.kr (M.J.Y.); sioh@jbnu.ac.kr (S.-I.O.); 2Laboratory for Infectious Disease Prevention, Korea Zoonosis Research Institute, Jeonbuk National University, 820-120, Hana-ro, Iksan 54531, Republic of Korea; tarkds@jbnu.ac.kr

**Keywords:** severe fever with thrombocytopenia syndrome (SFTS), co-house, transmission, mouse, interferon

## Abstract

Severe fever with thrombocytopenia syndrome (SFTS), a tick-borne zoonotic disease, is caused by infection with SFTS virus (SFTSV). A previous study reported that human-to-human direct transmission of SFTSV can occur. However, potential animal-to-animal transmission of SFTSV without ticks has not been fully clarified. Thus, the objective of this study was to investigate potential mice-to-mice transmission of SFTSV by co-housing three groups of mice [i.e., wild-type mice (WT), mice injected with an anti-type I interferon-α receptor-blocking antibody (IFNAR Ab), and mice with knockout of type I interferon-α receptor (IFNAR KO)] as spreaders or recipients with different immune competence. As a result, co-housed IFNAR Ab and IFNAR KO mice showed body weight loss with SFTS viral antigens detected in their sera, extracorporeal secretions, and various organs. Based on histopathology, white pulp atrophy in the spleen was observed in all co-housed mice except WT mice. These results obviously show that IFNAR Ab and IFNAR KO mice, as spreaders, exhibited higher transmissibility to co-housed mice than WT mice. Moreover, IFNAR KO mice, as recipients, were more susceptible to SFTSV infection than WT mice. These findings suggest that type I interferon signaling is a pivotal factor in mice intraspecies transmissibility of SFTSV in the absence of vectors such as ticks.

## 1. Introduction

Severe fever with thrombocytopenia syndrome (SFTS) is a tick-borne infectious disease. It is a widespread zoonotic disease in Asian countries, including China, Japan, Vietnam, Taiwan, and South Korea, causing a severe public health concern [1,2,3,4,5]. SFTS is caused by the SFTS virus (SFTSV), *Bandavirus dabieense*, which belongs to the genus *Bandavirus* in the family *Phenuiviridae* and the order *Bunyavirales* (ICTV, 2022). SFTSV infection can induce high fever, thrombocytopenia, leukocytopenia, digestive difficulties, and/or multiple organ dysfunction syndrome [6,7]. Owing to these symptoms, SFTS has a high mortality rate, particularly in the elderly with a higher death rate [8]. Immunity tends to decline with age, rendering elderly immunocompromised patients particularly susceptible to fatal outcomes [9]. A previous study using ferrets found that clinical symptoms, hematological analysis, and viral copy numbers associated with SFTS are higher in aged ferrets than in young adults [10]. Aging is associated with multiple defects in producing interferons in response to viral infections [11]. The antiviral state induced by interferon activity in virus-infected cells and tissues is mediated through the generation of interferon-stimulated genes [12]. Viruses can counteract these responses and diminish interferons’ antiviral efficacy by employing mechanisms that can facilitate their efficient replication. For example, tick-borne encephalitis virus, another tick-borne virus, can inhibit type 1 interferon signaling [13]. Wild-type mice infected with tick-borne encephalitis virus do not show clinical symptoms. However, mice with type 1 interferon receptor deficiency show clinical symptoms [14,15]. Likewise, a preliminary study on the pathogenicity of SFTS in mice has revealed that mice with regulated type 1 interferon signaling exhibit significantly higher pathogenicity than their wild-type counterparts [16].

The primary route of transmission for enzootic SFTSV is via ticks [17]. *Haemaphysalis longicornis* has been identified as a vector for SFTSV [18,19]. Although SFTS is known to be transmitted by ticks as a vector, a domestic cat infected with SFTS transmitting the virus to a human in Japan has been reported [20]. In addition, human-to-human transmission of SFTS cases has been reported in China and South Korea [21]. Previous studies on animal-to-animal transmission of SFTS have confirmed the possibility of its transmission in elderly ferrets and dogs [22,23]. In these studies, ferrets were capable of directly and indirectly transmitting SFTSV to others. In the case of dogs, direct transmission from SFTS-infected dogs to other dogs was observed. However, elderly ferrets and dogs are typically not easily accessible laboratory animals. Their limitations include high costs, difficulty of experimental operations, and a lack of immunoreagents that are readily available for purchase [24]. In contrast, mice offer several advantages, including ease of handling, diverse strains, and ease of immune modeling [24,25].

Based on this fact, we conducted experiments using wild-type (WT) mice, immunosuppressed mice injected with an anti-type I interferon-α receptor blocking antibody (IFNAR Ab), and immunocompromised mice with knockout of the type I interferon-α receptor (IFNAR KO). This study evaluated SFTSV transmissibility depending on type 1 interferon receptor signaling and compared the susceptibility of various immunocompromised mouse groups to SFTSV.

## 2. Materials and Methods

### 2.1. Ethics and Biosafety

All studies related to SFTSV infection were conducted in a Biosafety Level 3 (BL3) laboratory facility at the Korea Zoonosis Research Institute (KOZRI) at Jeonbuk National University. All experimental protocols underwent thorough review and received approval from the Animal Ethics Committee of Jeonbuk National University (approval no. JBNU 2021-011). Proper safety precautions were implemented when handling SFTSV.

### 2.2. Cell Culture and Viruses

The Korean SFTSV strain KH1 (GenBank accession No. MH491547, MH491548.1, and MH491549.1) was used in this study. This virus was passaged five times on monolayers of Vero E6 cells in Dulbecco’s Modified Eagle’s Medium (DMEM) containing 5% fetal bovine serum (FBS) with antibiotics (penicillin: 100 U/mL; streptomycin: 100 μg/mL) [16,23].

After plating the Vero cells onto 75 cm^2^ tissue culture flasks, they were exposed to the SFTSV strain. After an incubation period of five days at 37 °C, the cells underwent freezing and thawing three times. Subsequently, the harvested virus was titrated in 96-well microplates through 10-fold serial dilutions. Following this, the cells were fixed, stained with a specific NP hyperimmune serum against SFTSV, and then incubated with the alexafluor-488 anti-rabbit IgG (Invitrogen, Waltham, MA, USA) secondary antibody [26,27]. Virus infectivity titers were determined using a fluorescence-activated infectious dose (FAID_50_) assay. Mice were infected with a dose of 1.3 × 10^6^ FAID_50_ in a 100 μL volume of phosphate-buffered saline (PBS).

### 2.3. Infection of Animals and Sample Collection

Five-week-old female C57BL/6 mice (WT; Samtako, Kyoung Gi-do, Republic of Korea) and female IFNAR KO mice were acquired from B&K Universal (Hull, UK) and utilized in this study. To partially reduce type I IFN signaling, mice were intraperitoneally (IP) inoculated with 100 μg per mouse of the MAR1-5A3 (mouse anti-mouse IFNAR, IgG1, Bioxcell, Lebanon, NH, USA) monoclonal antibody (Mab) 1 day before SFTSV infection and 2 days post-infection (dpi). Infected mice were inoculated IP with SFTSV (KH1, P6, 1.3 × 10^6^ FAID_50_). 

Following infection, body weight was recorded daily. Blood was drawn from the retro-orbital venous plexus at 3, 6, and 9 dpi. Tears, saliva, and urine were collected at 2, 4, 6, and 8 dpi. All collected samples in 200 μL of PBS each were stored at −80 °C. At 5 and 10 dpi, both infected and co-housed groups were sacrificed, and organ tissues (lung, liver, kidney, spleen, and brain tissues) were collected. All collected samples were fixed in 10% neutral buffered formalin.

### 2.4. Experimental Design 

Experiments 1 and 2: To investigate the difference in the degree of transmission based on the spreader’s immunity, we used both WT and IFNAR KO mice as spreaders in the transmission experiment. As shown in Figure 1A,B, WT and IFNAR KO mice were inoculated with SFTSV (SFTSV-WT and SFTSV-KO; *n* = 12 each). IFNAR KO mice were co-housed with each inoculated group (Co-KO; *n* = 12 each). Six mice of the inoculated group and six mice of the co-housed group were housed in individual cages (2 cages for each group).

Experiment 3: To compare the difference in susceptibility to SFTSV depending on the recipient mice’s immunity, we used WT, IFNAR Ab, and IFNAR KO mice as recipients. As spreaders, IFNAR Ab mice were inoculated with SFTSV (SFTSV-Ab; *n* = 12 each) (Figure 1C). WT, IFNAR Ab, and IFNAR KO mice (Co-WT, Co-Ab, and Co-KO; *n* = 12 each) were individually placed in distinct cages with SFTSV-Ab mice. In each cage, there were six mice from the inoculated group and six mice from the co-housed group (2 cages for each group).

### 2.5. Quantitative Polymerase Chain Reaction (qPCR) for Detection of SFTSV RNA

Hybrid-R™ (GeneAll, Seoul, Republic of Korea) was utilized for isolating total RNA. RNA underwent reverse transcription to generate cDNA using a ReverTra Ace qPCR RT Master Mix (TOYOBO, Osaka, Japan). qPCR with an L segment-based SFTSV-specific primer was employed to determine viral copy levels. The forward primer sequence was SFTSV-L-F: AACATCCTGGACCTTGCATC, and the reverse primer sequence was SFTSV-L-R: CAATGTGGCCATCTTCTCCA [28]. Copy numbers were calculated by comparing them with a standard control. A CFX96 Real-time PCR system (Bio-Rad Laboratories, Contra Costa, CA, USA) was used in conjunction with a qPCR SyGreen Mix (PCR Biosystems, Seoul, Republic of Korea).

### 2.6. Microscopic Observation

Tissues preserved in 10% neutral buffered formalin underwent routine processing using a Myr Spin tissue processor-STP 120 (Myr, Tarragona, Spain). Employing an embedding station (Sakura, San Ramon, CA, USA), processed tissues were embedded in paraffin wax (Leica Biosystems, Milton Keynes, Bu, UK). Tissue sections (5 µm in thickness) were obtained using a microtome craftex CR-603 (Leica Biosystems) and mounted on glass slides (Muto pure chemicals, Tokyo, Japan). Hematoxylin and eosin (H&E) staining was performed automatically using a Myr Eva-slide stainer SS-30 (Myr). A silane-coated slide (Muto pure chemicals) was utilized for immunohistochemistry (IHC). We employed a citrate buffer (pH 6.0) at 95 °C for 30 min and at room temperature for 20 min to restore immunoactivity. Tissue slides were incubated with an SFTSV N protein antibody (provided by Dr. Jun-Gu Choi, the Animal and Plant Quarantine Agency, Gimcheon, the Republic of Korea) as the primary antibody at 4 °C overnight. The primary antibody was labeled with horseradish peroxidase-conjugated anti-mouse/rabbit immunoglobulin G antibody (MP-7500, Vector Laboratories, Newark, CA, USA). To visualize the antibody, 3.3′-diaminobenzidine (SK-4105, Vector Laboratories) was employed at a dilution of 1:30. For counterstaining, slides were incubated with methyl green for 10 min. An Olympus microscope (BX53F; Olympus, Tokyo, Japan) and digital imaging software (Olympus) were used for digital imaging.

### 2.7. Statistical Analysis

Statistically significant differences were analyzed using an unpaired *t*-test or Mann–Whitney test with GraphPad Prism version 8.00. Asterisks were used to indicate statistical significance in respective graphs (****, *p* < 0.0001; ***, *p* < 0.001; **, *p* < 0.01; and *, *p* < 0.05). 

## 3. Results

### 3.1. Experiment 1: Confirmation of SFTSV Transmission Capability from WT Mice to Immunocompromised Mice

We investigated the potential transmission of SFTSV to IFNAR KO mice, which are most vulnerable to SFTSV, as recipients, aiming to evaluate the virus transmission capability (Figure 1A).

During the entire experimental period, there were no fatalities of SFTSV-WT or Co-KO mice due to SFTSV infection (Figure 2A). Both groups showed slight body weight loss until 2 dpi (Appendix A). Despite a low viral burden, SFTSV viral loads were detected in the sera at 3 dpi in almost all experimental mice. Furthermore, at 3 dpi, one Co-KO mouse had the highest viral load (137 copies/100 ng RNA) in its serum (Figure 2B). From 2 dpi, the majority of SFTSV-WT mice shed SFTSV via tears, saliva, and urine. The Co-KO group shed SFTSV through all types of analyzed specimens at 2, 4, and 6 dpi. In all swab samples at 4 dpi, the mean SFTS viral loads in the Co-KO group were significantly higher than those in the SFTSV-WT group. As shown in Figure 2C, SFTSV viral loads were detected in at least one mouse from each group in all organs at 5 dpi. The average viral load in the spleen in the Co-KO group was significantly higher (*p* = 0.0108) at 10 dpi than at 5 dpi. It was similar to the viral load detected in the SFTSV-WT group. 

Histopathological analysis of the Co-KO mice livers revealed multifocal mild inflammatory foci, primarily consisting of mononuclear cells (Figure 2D). White pulp atrophy lesions known to be characteristic lesions of SFTS were observed in the spleens of Co-KO mice at 5 and 10 dpi [29]. In addition, SFTSV nucleocapsid protein (NP) antigen-positive cells were primarily detected near the white pulp of the spleen at both 5 and 10 dpi, especially at 10 dpi (Figure 2E). These results indicate that the virus can be transmitted from normal mice infected with SFTSV to immunocompromised mice such as IFNAR KO mice.

### 3.2. Experiment 2: Higher SFTSV Transmissibility in IFNAR KO Mice Than in WT Mice

Through Experiment 1, we validated the transmission of SFTSV from WT mice to IFNAR KO mice. To determine the transmission efficiency of SFTSV depending on the immune levels of spreaders, we designed Experiment 2 using SFTSV-infected INFAR KO mice as spreaders (Figure 1B). 

SFTSV-KO mice showed a mortality rate of 100% at 4 dpi, while Co-KO mice began to die at 7 dpi and ultimately showed a mortality rate of 50% (Figure 3A). The SFTSV-KO group exhibited a steady body weight loss, reaching 80% until death, while the surviving Co-KO group maintained their body weight (Appendix A). In the case of fatal Co-KO mice, they experienced extreme body weight losses of over 20% until death. As shown in Figure 3B, although an SFTS viral load was detected in the serum of the Co-KO group, its levels were comparatively lower than that detected at 3 dpi in the SFTSV-KO group. For tear and urine samples from the Co-KO group, an SFTS viral load was detected in only one mouse at 8 dpi. The viral titer was higher than that detected in the urine of the SFTSV-KO group at 2 dpi. An SFTS viral load in the saliva was detected only in the SFTSV-KO group at 2 dpi.

Necropsies were conducted for the Co-KO group at 5 and 10 dpi. The organs of fatal mice were harvested on the date of death. In fatal mice from the SFTSV-KO and Co-KO groups, high SFTS viral titers were detected in all organs (Figure 3C). They showed no statistically significant differences between fatal SFTSV-KO mice and fatal Co-KO mice. Interestingly, an SFTS viral load was detected in the spleen of the surviving Co-KO mouse at 10 dpi.

As shown in Figure 3D, multifocal inflammatory foci primarily consisting of mononuclear cells were observed in the livers of mice in the Co-KO group based on H&E staining (arrows). In fatal mice of the Co-KO group, perivascular mononuclear inflammatory cell infiltration was observed (arrowhead). White pulp atrophy lesions were observed in spleens at 5 dpi and in fatal Co-KO mice (asterisks). Additionally, diminished cellularity was observed in the red pulp of all Co-KO group mice. Pathological changes in the spleen described above were observed more severely in fatal Co-KO mice than in surviving Co-KO mice. Furthermore, necrotic lesions were observed in the spleens of fatal Co-KO mice. SFTSV NP antigen-positive cells were noted near the white pulp of spleens of all Co-KO mice (Figure 3E). These results indicate that SFTSV transmissibility is higher in immunocompromised mice such as IFNAR KO mice than in WT mice.

### 3.3. Experiment 3: IFNAR KO Mice Have Higher Susceptibility to SFTSV Than WT Mice

Through the preceding two experiments, we demonstrated the possibility of SFTSV transmission from mice to mice, indicating that this virus could spread more efficiently between immunocompromised mice. In the third experiment, we assigned mice with various forms of type 1 interferon receptor signaling as recipients to investigate the role of type 1 interferon receptors in SFTSV infection (Figure 1C).

Mice in the Co-WT group did not show any clinical signs during the experiment (Figure 4A). One mouse each from the Co-Ab and Co-KO groups died at 1 dpi. Among mice in the co-housed group, only mice in the Co-KO group showed clinical symptoms accompanied by body weight loss in the middle of this study (Appendix A).

In the serum samples, SFTS viral loads in mice of the Co-Ab and Co-KO groups were observed only at 9 dpi (Figure 4B). The viral titer results show no statistically significant difference between the co-housed and SFTS-Ab groups at 9 dpi. Interestingly, an SFTS viral load was detected in all types of swabs (tears, saliva, and urine) obtained from almost all co-housed groups. In particular, in the saliva samples, the SFTS viral load in the Co-KO group showed a statistically significant difference from the SFTSV-Ab group at 2 dpi. However, it increased at 8 dpi to a level that showed no significant difference from that in the SFTSV-Ab group.

As shown in Figure 4C, SFTSV was detected in all organs collected from all groups at 5 dpi. The co-housed groups had higher SFTS viral copy numbers in the lungs, livers, and brains than the SFTSV-Ab group, although such differences were not statistically significant. The mean numbers of SFTS viral copies in spleens were comparable between the SFTS-Ab and co-housed groups. The SFTS viral load in the spleens of the co-housed group showed an increase at 10 dpi compared with that at 5 dpi, although the number of SFTSV-positive animals was low. Especially for the Co-Ab and Co-KO groups, SFTS viral loads were generally detected in the organs of one or two mice. Their SFTS viral loads were mostly the highest in each organ examined. However, statistical analysis could not be performed due to an insufficient number of mice.

As a result of histopathological examination (Figure 4D), mild inflammatory lesions were observed in the livers of mice in the Co-WT and Co-Ab groups at 5 dpi (arrows). The livers of Co-KO-group mice showed severe inflammatory cell infiltration and foci of necrotic lesions (crosshatches). White pulp atrophy lesions were observed only in the spleens of mice in the Co-KO group (asterisks). SFTSV NP antigen-positive cells were observed in all groups, with the Co-Ab and Co-KO groups showing higher reactivity than the Co-WT group. At 10 dpi (Figure 4E), a mild inflammatory lesion was observed in the livers of mice in the Co-Ab group (arrow), while severe multifocal inflammatory cell infiltration and foci of necrotic lesions were observed in the livers of mice in the Co-KO group (crosshatches). White pulp atrophy lesions were observed in the spleens of mice in both the Co-Ab and Co-KO groups (asterisks). SFTSV NP antigen-positive cells were present in all groups, with the Co-KO group showing the highest reactivity. These results indicate that IFNAR KO mice have the highest SFTSV susceptibility and that IFNAR Ab is suitable as an SFTSV spreader.

## 4. Discussion

This study examined the transmission effectiveness of directly inoculating a spreader group of mice using IFNAR KO mice as recipients. The results show that the SFTS viral loads in co-housed mice were greater than in inoculated mice for all collected samples at 4 dpi (Figure 2B). This meant that SFTSV was successfully transmitted from mice to mice. Moreover, individuals with a secondary infection might have contributed to a more extensive transmission through secretions. A previous study demonstrated that the secondary infectivity of parainfluenza virus and rhinovirus might be stronger than the primary infectivity [30]. Therefore, not only animals directly infected with SFTSV but also animals with secondary infection might have a significantly high risk of transmitting the virus to co-housed animals with an immune-compromised situation. 

We evaluated the differences in virus transmissibility based on the presence or absence of IFNAR signaling in the spreader. In experiment 2, the Co-KO group showed severe clinical and histopathological symptoms, and 50% of mice in the Co-KO group succumbed to infection. Notably, mice in the Co-KO group began to die after all mice in the SFTSV-KO group had died. The deaths of mice in the Co-KO group in experiment 2 were presumed to be caused by SFTSV secreted by mice in the SFTSV-KO group that resided in the same cage. Transmission of SFTSV from a critically ill SFTS patient to humans through non-blood contact has also been reported [31]. Although there was no direct contact with the critically ill SFTS patient, the spread of the virus was confirmed through the SFTSV secreted by the critically ill SFTS patient remaining in the same space. In addition, previous papers have shown that immunocompromised patients are more susceptible to the virus than normal patients [32,33,34]. Similarly, the present study revealed that immunocompromised mice without IFNAR signaling showed higher susceptibility to SFTSV than normal mice.

We investigated the SFTSV susceptibilities of co-housed groups with different immunities. Among the three recipient groups, only the Co-KO group showed body weight loss. Similar SFTS viral copy numbers were detected in serum samples of the Co-Ab and Co-KO groups of mice at 9 dpi. Previous research data showed that SFTSV has the capability to disrupt the initial induction of type I IFNs by targeting host kinases TBK1/IKKε [35,36,37]. Additionally, SFTSV has been observed to inhibit type I IFN-triggered signaling pathways and interferon-stimulated gene expression [38]. These results showed that, as with the direct inoculation of SFTSV in mice, the susceptibility of co-housed mice to SFTSV increased when type 1 interferon signaling decreased [39]. Furthermore, the SFTS viral titers in the extracorporeal secretions of the Co-WT group were similar to those of the SFTSV-Ab group. However, like wildlife rodents [40], clinical signs were rarely observed, and SFTS viral copy numbers were hardly detected in the sera. Similar to previous studies indicating the possibility of viral shedding even in asymptomatic patients [41], these findings suggest that SFTSV infection occurs even in normal mice without apparent clinical signs and that transmission is possible through extracorporeal secretions. Viral loads from swab samples in co-housed groups were detected to be higher at 8 dpi than at 6 dpi. These results suggest that cross-infection between mice occurred. Similar to other infectious diseases with the risk of secondary infection increasing in confined spaces, SFTSV also showed possible cross-infection through close contact in crowded environments [42]. Considering this, when wild mice, which have the characteristic of social organization, are infected with SFTSV, the possibility of the virus spreading may be increased [43]. 

From a histopathological standpoint, the spleen, liver, and kidney exhibit lesions upon SFTSV infection [44]. Infiltration of inflammatory cells and necrotic lesions are observed in the liver. In the spleen, atrophy of the white pulp is typically observed. In severe cases, a decrease in the cellularity of red pulp is also observed [45]. In immunohistochemistry, macrophages that phagocytose SFTSV NP are detected around the white pulp of the spleen. As shown in Figure 3 and Figure 4, atrophy of the white pulp of the spleen and a decrease in the cellularity of red pulp were also observed in Co-KO mice. In addition, SFTSV NP antigen-positive cells were observed around the white pulp based on IHC. In Figure 3E, surprisingly, the number of SFTSV NP antigen-positive cells in the spleens of fatal Co-KO mice, which had the highest SFTS viral load, was less than that in spleens at 5 or 10 dpi. In dead Co-KO mice, the fewer SFTSV antigen-positive cells observed might be because multiple organ dysfunction occurred, and problems occurred in the phagocytic pathway of macrophages, resulting in poor phagocytosis [46]. In experiment 3, upon histological examination, lesions were observed more severely in the following order: the Co-WT, Co-Ab, and Co-KO groups (Figure 4D,E). Local necrosis and infiltration of inflammatory cells were evident in the livers and white pulp atrophy was observed in the spleens in the Co-KO group, with more SFTSV NP antigen-positive cells detected compared with other groups. Interestingly, an inflammatory reaction in the kidney, a histopathological characteristic of SFTSV infection [47], was observed in the inoculated group but not in the co-housed groups. Because there could be a difference in pathogenicity between individuals infected by direct injection and those infected through transmission [48], it was expected that no kidney lesions would appear in the co-housed group. Our results suggest that SFTSV can be transmitted to animals through secretions and that these animals infected through secretion exhibit histological lesions similar to those of directly infected individuals. Additionally, in situations where the virus is naturally transmitted rather than experimentally inoculated, a lower level of type I interferon signaling appears to correlate with higher susceptibility.

Ticks, as important vectors of SFTSV, could be responsible for the virus’s vertical transmission according to previous studies [49,50]. Additionally, although no cases of SFTS transmission through sexual contact have been reported in humans, the detection of SFTSV RNA in semen suggests possible transmission through sexual contact [51]. As a limitation of our study, we only confirmed the possibility of horizontal intra-species transmission in mice. Based on these results, we believe that further research is needed to investigate vertical intra-species transmission as well as transmission through sexual contact. 

In SFTSV transmission experiments, IFNAR Ab mice might have performed better as spreaders than WT and IFNAR KO mice. Despite having a lower mortality rate and higher SFTS viral loads than WT mice, IFNAR Ab mice offered the advantage of allowing experiments to be designed for longer durations. IFNAR KO mice are most suitable as recipients in SFTSV transmission experiments because they not only present severe clinical signs but also show serious histopathological lesions. Our findings demonstrate that it is an ideal mouse model for transmission experiments of SFTSV. These models can be used to investigate SFTSV transmission and study transmission vectors in the absence of ticks. Type I interferon signaling may be a pivotal factor for the transmissibility of SFTSV without a vector such as a tick. Further detailed pathogenicity and transmissibility studies are needed to reveal the role of carriers in SFTSV infection.

## Figures and Tables

**Figure 1 viruses-16-00401-f001:**
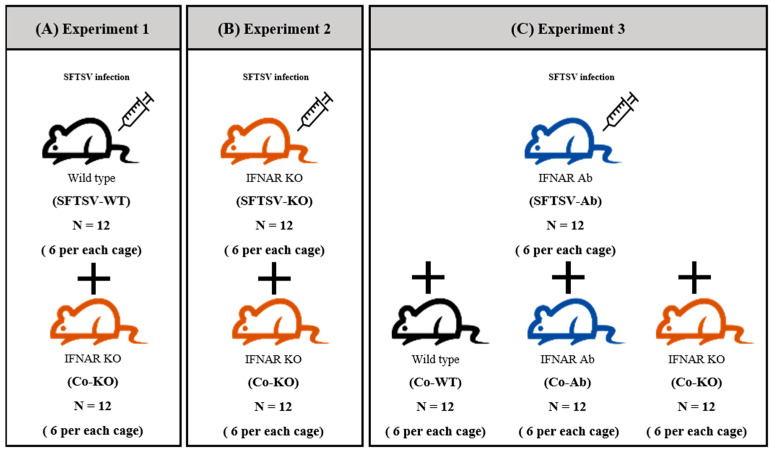
Experimental design. WT (Black), IFNAR Ab (Blue), and IFNAR KO (Red) mice served as spreader and recipient groups, respectively. (**A**–**C**) Spreader group mice were IP inoculated with SFTSV (KH1; 1.3 × 10^6^ FAID_50_). Each group had two cages. Six mice of the inoculated group and six mice of the co-housed group were housed in individual cages (2 cages for each group).

**Figure 2 viruses-16-00401-f002:**
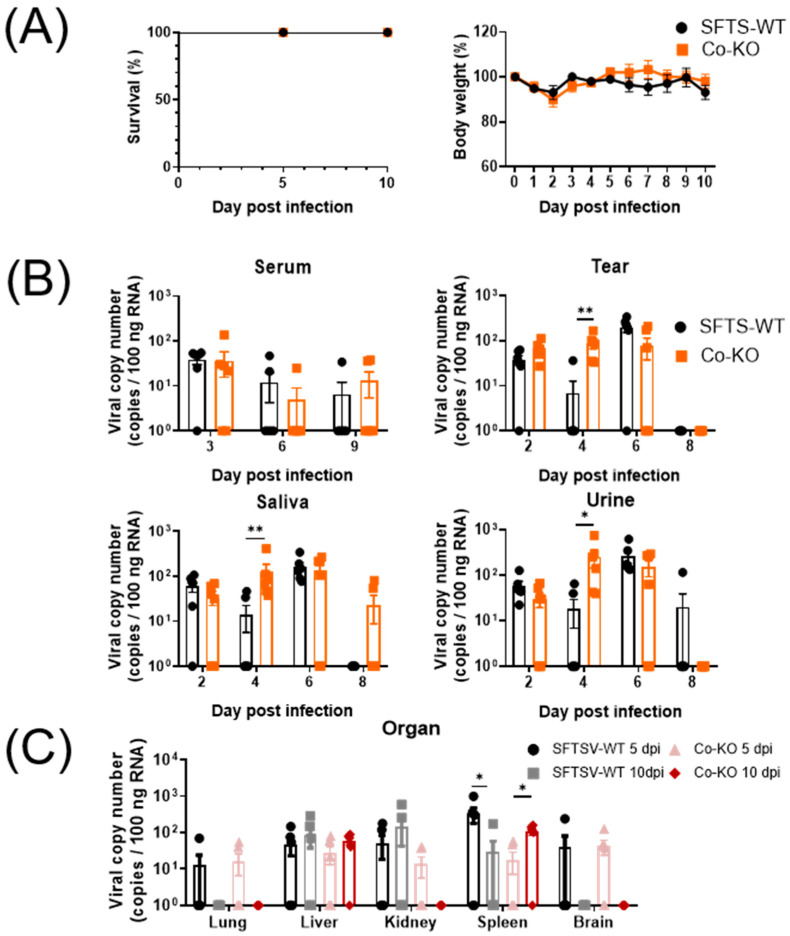
Confirmation of SFTSV transmission from WT mice to immunocompromised mice. (**A**) Survival rate and body weight are presented as percentages. (**B**) Viral load detected in sera, tears, saliva, and urine obtained in SFTSV-WT and Co-KO groups. (**C**) STFS viral qPCR results. (**D**) Black arrows indicate moderate multifocal inflammatory foci in the liver, and asterisks indicate white pulp atrophy in the spleen. H&E staining scale bar: 100 μm and 500 μm. (**E**) SFTSV NP antigen-positive cells were detected around the white pulp in the spleen. IHC staining scale bar: 100 μm and 50 μm. Data are expressed as means ± SEM. Statistical analysis: An unpaired *t*-test was used to compare each group’s body weight. Mann–Whitney test was used to compare SFTS viral copy number: **, *p* < 0.01; *, *p* < 0.05.

**Figure 3 viruses-16-00401-f003:**
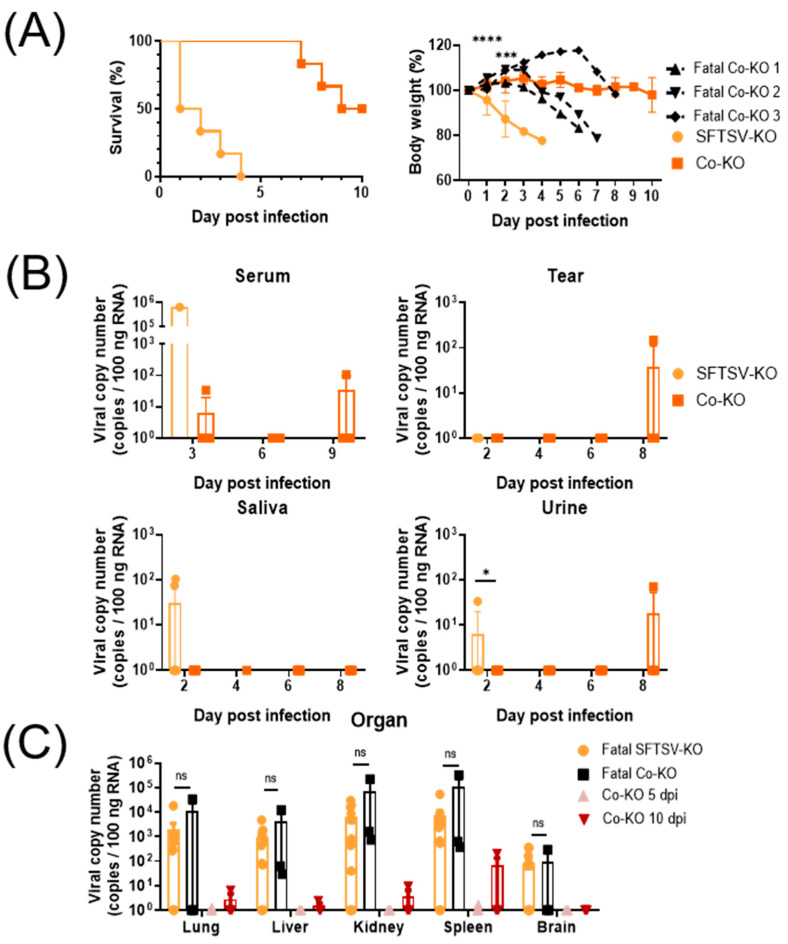
Higher SFTSV transmissibility in IFNAR KO mice than in WT mice. (**A**) Survival rate and body weight are presented as percentages. (**B**) Viral load detected in sera, tears, saliva, and urine obtained from SFTSV-KO and Co-KO mice. (**C**) STFS viral qPCR. (**D**) Black arrows indicate mild multifocal inflammatory foci in the liver. Arrowhead indicates perivascular mononuclear inflammatory cell infiltration in the liver. Asterisks indicate white pulp atrophy in the spleen. Box indicates focal necrosis area in the spleen of a fatal case. H&E staining scale bar: 100 μm and 500 μm. (**E**) SFTSV NP antigen-positive cells were detected around the white pulp in the spleen. IHC staining scale bar: 100 μm and 50 μm. Data are expressed as means ± SEM. Statistical analysis: An unpaired *t*-test was used to compare each group’s body weight. Mann–Whitney test was used to compare SFTS viral copy number: ****, *p* < 0.0001; ***, *p* < 0.001; and *, *p* < 0.05.

**Figure 4 viruses-16-00401-f004:**
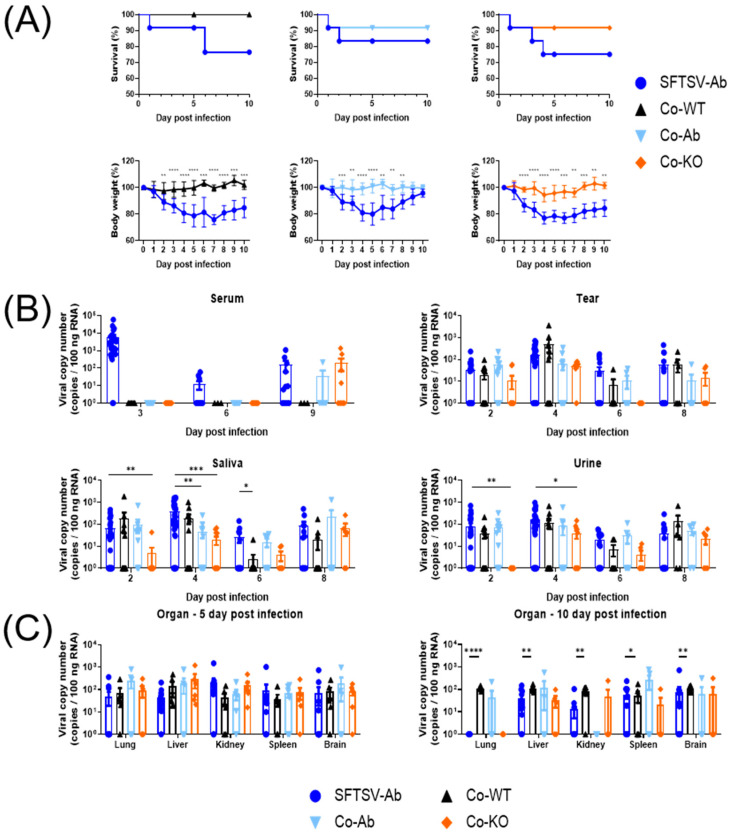
IFNAR KO mice have higher susceptibility to SFTSV than WT mice. (**A**) Survival rate and body weight are presented as percentages. (**B**) Viral loads were detected in sera, tears, saliva, and urine obtained from SFTSV-Ab group and three co-housed mice groups. (**C**) SFTS viral qPCR. (**D**) Black arrows indicate mild multifocal inflammatory foci in the liver. Crosshatch indicates focal necrosis area in the liver. Asterisks indicate white pulp atrophy in the spleen. SFTSV NP antigen-positive cells were detected around the white pulp in the spleen. H&E staining scale bar: 100 μm and 500 μm. IHC staining scale bar: 100 μm. (**E**) Black arrow indicates minor multifocal inflammatory foci in the liver. Crosshatches in the liver suggest a focal necrosis region. Asterisks indicate splenic white pulp atrophy. H&E staining scale bar: 100 μm and 500 μm. IHC staining scale bar: 100 μm. Data are expressed as means  ±  SEM. Statistical analysis: An unpaired *t*-test was used to compare each group’s body weight. Mann–Whitney test was used to compare SFTS viral copy number: ****, *p* < 0.0001; ***, *p* < 0.001; **, *p* < 0.01; and *, *p* < 0.05.

## Data Availability

The data presented in this study are available upon request from the corresponding authors.

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
