# Peer review of "Difference in Intraspecies Transmissibility of Severe Fever with Thrombocytopenia Syndrome Virus Depending on Abrogating Type 1 Interferon Signaling in Mice"

_viruses, 2024, doi:10.3390/v16030401_

Round 1
Reviewer 1 Report
Comments and Suggestions for Authors
Comments on Oh et al “Difference of intraspecies transmissibility of severe fever with thrombocytopenia syndrome virus depending on mice regulating type 1 interferon signaling”
This is an interesting paper that documents animal-to-animal spread of SFTSV under controlled conditions. It is an important area to study, since it has implications for human health in endemic areas. Overall the paper is very straightforward in design, and the results are worthy of publication. However, the paper requires some modifications before it is ready.
Title: I don’t think “…depending on mice regulating type 1 interferon signaling” is a good description, as it sounds like the ‘regulation’ would be part of a normal response to viral infection. Here, the interferon pathway has been artificially abrogated in two different ways, and so the main point is that abrogation of the interferon pathway (in any way) enhances animal to animal spread. Title should be modified to more accurately reflect the finding.
More description and a reference should be added to the FAID50 assay in the Materials and Methods
Figure 1 and text: it is not clear how many of each kind of mice are housed in individual cages, this should be made more explicit. Also, in the figure the font is too small to read.
Figure 2, 3, 4 (and text in Results): it is interesting to see the averages in the graphs, but in this case it is important to see the results of all of the mice. As presented, it is hard to tell how many of each kind of mouse got high or low viral loads, etc. Since this is a first of kind study, it is important to know the inter-individual variability. Perhaps some of these data could be presented as a table and summarized in graphs as averages. Also, these figures are far too small to read as is.
In the discussion, it is very interesting to note that the effects of administration of the IFNAR Ab give subtly different results than the KO of IFNARa. This should be discussed and interpreted at greater length.
Comments on the Quality of English Language
Although the text is intelligible, there are some awkward sentences, verb-noun agreement issues, and a few typos that would benefit from proofreading by a native English speaker.
Author Response
<<Cover letter>>
We would like to thank Editors and Reviewers for careful evaluation on our manuscript. We agree the reviewers’ comments are insightful and their suggestions are excellent. We have revised the manuscript extensively as suggested by the reviewers. We hope this revised manuscript is suitable for publication in viruses. Thank you for your consideration of our manuscript. We look forward to hearing your and reviewers’ comments. The following is our point by point response to the reviewer’s comments.
<<Point by point response>>
Comments on Oh et al “Difference of intraspecies transmissibility of severe fever with thrombocytopenia syndrome virus depending on mice regulating type 1 interferon signaling”
This is an interesting paper that documents animal-to-animal spread of SFTSV under controlled conditions. It is an important area to study, since it has implications for human health in endemic areas. Overall the paper is very straightforward in design, and the results are worthy of publication. However, the paper requires some modifications before it is ready.
- Title: I don’t think “…depending on mice regulating type 1 interferon signaling” is a good description, as it sounds like the ‘regulation’ would be part of a normal response to viral infection. Here, the interferon pathway has been artificially abrogated in two different ways, and so the main point is that abrogation of the interferon pathway (in any way) enhances animal to animal spread. Title should be modified to more accurately reflect the finding.
Thank you for pointing this out. As suggested by reviewer’s comments, we modified “regulating” to “abrogating” in title. As the reviewer said, ‘regulating’ seems to be a word that can be easily misunderstood to mean that the mouse regulates itself through an immune response. Therefore, we modified the wording from ‘regulating’ to ‘abrogating’ to match our experimental .
- More description and a reference should be added to the FAID50 assay in the Materials and Methods
We thank the reviewer for bringing up this point. As suggested by reviewer’s comments, we included more description and references in Materials and Methods (Page 2, line 81-89).
- Figure 1 and text: it is not clear how many of each kind of mice are housed in individual cages, this should be made more explicit. Also, in the figure the font is too small to read.
We thank the reviewer for bringing up this point. As suggested by reviewer’s comments, we added more explanation in page 3, line 111-113. Additionally, we added 'in the one cage' to clarify the number of mice per cage in Figure 1 legend (Page 3, line 125-126). Also, we enlarged the font size of figure 1.
- Figure 2, 3, 4 (and text in Results): it is interesting to see the averages in the graphs, but in this case it is important to see the results of all of the mice. As presented, it is hard to tell how many of each kind of mouse got high or low viral loads, etc. Since this is a first of kind study, it is important to know the inter-individual variability. Perhaps some of these data could be presented as a table and summarized in graphs as averages. Also, these figures are far too small to read as is.
Thank you for pointing this out. We had already displayed individual mouse values (although many values were overlapped) in all graphs except the weight graph. As suggested reviewer’s comments, it is also important to know the amount of change for each individuals. Therefore, we will tabulate the body weight change of each group and provide it as supplementary data. Also, we enlarged the size of figure (Page 5, 7, and 9).
- In the discussion, it is very interesting to note that the effects of administration of the IFNAR Ab give subtly different results than the KO of IFNARa. This should be discussed and interpreted at greater length.
Thank you for pointing this out. Previous research data showed that SFTSV has the capability to disrupt the initial induction of type I IFNs by targeting host kinases TBK1/IKKε [1-3]. Additionally, SFTSV has been observed to inhibit type I IFN-triggered signaling pathways and interferon stimulated gene expression [4]. Additionally, in a previous paper, differences in pathogenicity according to interferon signals were compared when mice were inoculated with viruses [5]. The contents were added to the discussion (Page 10, line 322-327).
<Reference>
- Ning, Y.-J., et al., Viral suppression of innate immunity via spatial isolation of TBK1/IKKε from mitochondrial antiviral platform. Journal of molecular cell biology, 2014. 6(4): p. 324-337.
- Wu, X., et al., Evasion of antiviral immunity through sequestering of TBK1/IKKε/IRF3 into viral inclusion bodies. Journal of virology, 2014. 88(6): p. 3067-3076.
- Santiago, F.W., et al., Hijacking of RIG-I signaling proteins into virus-induced cytoplasmic structures correlates with the inhibition of type I interferon responses. Journal of virology, 2014. 88(8): p. 4572-4585.
- Ning, Y.-J., et al., Disruption of type I interferon signaling by the nonstructural protein of severe fever with thrombocytopenia syndrome virus via the hijacking of STAT2 and STAT1 into inclusion bodies. Journal of virology, 2015. 89(8): p. 4227-4236.
- Park, S.C., et al., Pathogenicity of severe fever with thrombocytopenia syndrome virus in mice regulated in type I interferon signaling: Severe fever with thrombocytopenia and type I interferon. Lab Anim Res, 2020. 36: p. 38.
<<Supplementary data>>
1. Body weight change
| dpi NAME |
0 | 1 | 2 | 3 | 4 | 5 | 6 | 7 | 8 | 9 | 10 |
| Experiment 1 |   |   |   |   |   |   |   |   |   |   |   |
| SFTSV-WT 1 | 100 | 91.74 | 103.42 | 98.32 | 96.77 | 99.75 |   |   |   |   |   |
| SFTSV-WT 2 | 100 | 90.74 | 83.7 | 98.4 | 99.32 | 99.81 |   |   |   |   |   |
| SFTSV-WT 3 | 100 | 90.61 | 82.68 | 95.98 | 94.57 | 96.95 |   |   |   |   |   |
| SFTSV-WT 4 | 100 | 88.23 | 80.61 | 101.52 | 98.78 | 99.63 |   |   |   |   |   |
| SFTSV-WT 5 | 100 | 88.81 | 78.75 | 87.75 | 92 | 92.19 |   |   |   |   |   |
| SFTSV-WT 6 | 100 | 90.7 | 82.8 | 103.63 | 101.4 | 103.63 |   |   |   |   |   |
| SFTSV-WT 7 | 100 | 93.46 | 94.65 | 96.04 | 93.9 | 93.27 | 85.28 | 80.69 | 79.56 | 81.38 | 78.93 |
| SFTSV-WT 8 | 100 | 100.06 | 102.3 | 103.03 | 103.45 | 107.15 | 106.97 | 105.45 | 106.3 | 104.73 | 100.48 |
| SFTSV-WT 9 | 100 | 98.7 | 98.39 | 99.75 | 96.89 | 101.12 | 100.62 | 98.76 | 103.6 | 101.86 | 93.54 |
| SFTSV-WT 10 | 100 | 101.11 | 102.9 | 103.27 | 100 | 92.78 | 92.59 | 94.01 | 95.37 | 95.68 | 93.4 |
| SFTSV-WT 11 | 100 | 100.06 | 99.52 | 102.57 | 95.33 | 94.97 | 93.23 | 92.99 | 95.39 | 110.48 | 92.4 |
| SFTSV-WT 12 | 100 | 104.05 | 107.59 | 110.38 | 103.92 | 106.65 | 100.13 | 100.63 | 102.41 | 104.24 | 100.13 |
|   |   |   |   |   |   |   |   |   |   |   |   |
| Co-KO 1 | 100 | 87.99 | 78.49 | 96.04 | 99.28 | 104.89 |   |   |   |   |   |
| Co-KO 2 | 100 | 86.84 | 77.61 | 96.06 | 95.16 | 103.48 |   |   |   |   |   |
| Co-KO 3 | 100 | 87.89 | 79.1 | 95.36 | 98.86 | 103.98 |   |   |   |   |   |
| Co-KO 4 | 100 | 88.42 | 80.25 | 93.35 | 94.62 | 99.05 |   |   |   |   |   |
| Co-KO 5 | 100 | 88.72 | 80.99 | 72.27 | 104.65 | 107.15 |   |   |   |   |   |
| Co-KO 6 | 100 | 89.84 | 81.64 | 97.43 | 96.72 | 97.6 |   |   |   |   |   |
| Co-KO 7 | 100 | 103.5 | 99.49 | 99.27 | 93.72 | 98.39 | 99.27 | 100.15 | 96.2 | 96.35 | 94.89 |
| Co-KO 8 | 100 | 102.14 | 102.74 | 105.89 | 104.29 | 115.48 | 113.1 | 114.46 | 105.71 | 108.93 | 107.86 |
| Co-KO 9 | 100 | 107.57 | 104.8 | 103.62 | 98.88 | 108.55 | 107.5 | 110 | 102.96 | 98.03 | 93.62 |
| Co-KO 10 | 100 | 99.42 | 99.06 | 99.42 | 99.77 | 101.81 | 105.26 | 108.77 | 109.77 | 107.66 | 106.67 |
| Co-KO 11 | 100 | 101.45 | 91.56 | 90.06 | 85.32 | 85.32 | 87.75 | 87.75 | 89.6 | 91.85 | 89.36 |
| Co-KO 12 | 100 | 103.24 | 101.9 | 101.69 | 96.97 | 98.8 | 99.08 | 98.59 | 96.27 | 96.83 | 95.92 |
|   |   |   |   |   |   |   |   |   |   |   |   |
|   |   |   |   |   |   |   |   |   |   |   |   |
| Experiment 2 |   |   |   |   |   |   |   |   |   |   |   |
| SFTSV-KO 1 | 100 | 95.19 | 84.49 |   |   |   |   |   |   |   |   |
| SFTSV-KO 2 | 100 | 96.65 | 88.83 | 77.99 |   |   |   |   |   |   |   |
| SFTSV-KO 3 | 100 | 92.05 | 86.59 |   |   |   |   |   |   |   |   |
| SFTSV-KO 4 | 100 | 97.92 |   |   |   |   |   |   |   |   |   |
| SFTSV-KO 5 | 100 | 93.98 | 83.73 |   |   |   |   |   |   |   |   |
| SFTSV-KO 6 | 100 | 92.43 | 80.76 |   |   |   |   |   |   |   |   |
| SFTSV-KO 7 | 100 | 89.81 |   |   |   |   |   |   |   |   |   |
| SFTSV-KO 8 | 100 |   |   |   |   |   |   |   |   |   |   |
| SFTSV-KO 9 | 100 | 102.72 | 92.93 | 81.79 | 77.72 |   |   |   |   |   |   |
| SFTSV-KO 10 | 100 |   |   |   |   |   |   |   |   |   |   |
| SFTSV-KO 11 | 100 |   |   |   |   |   |   |   |   |   |   |
| SFTSV-KO 12 | 100 | 94.51 | 81.59 |   |   |   |   |   |   |   |   |
|   |   |   |   |   |   |   |   |   |   |   |   |
| Co-KO 1 | 100 | 97.06 | 98.82 | 107.53 | 109.35 | 104.71 |   |   |   |   |   |
| Co-KO 2 | 100 | 88.3 | 89.47 | 98.01 | 93.8 | 87.72 |   |   |   |   |   |
| Co-KO 3 | 100 | 101.66 | 101.66 | 87.18 | 84.7 | 83.43 |   |   |   |   |   |
| Co-KO 4 | 100 | 107.66 | 105.53 | 107.57 | 106.64 | 104.17 |   |   |   |   |   |
| Co-KO 5 | 100 | 106.02 | 105.42 | 102.95 | 101.69 | 102.41 |   |   |   |   |   |
| Co-KO 6 | 100 | 86.01 | 89.51 | 96.5 | 101.33 | 106.29 |   |   |   |   |   |
| Co-KO 7 | 100 | 102.31 | 98.73 | 105.09 | 101.73 | 101.73 | 102.2 | 98.61 | 100.12 | 100.69 | 100.23 |
| Co-KO 8 | 100 | 99.49 | 104.39 | 102.96 | 99.9 | 103.88 | 99.44 | 98.98 | 98.21 | 101.17 | 89.39 |
| Co-KO 9 | 100 | 100.48 | 108.93 | 112.44 | 115.77 | 117.2 | 117.8 | 108.27 | 98.33 |   |   |
| Co-KO 10 | 100 | 105.17 | 109.2 | 108.22 | 106.44 | 108.22 | 101.95 | 102.53 | 106.32 | 102.87 | 104.25 |
| Co-KO 11 | 100 | 105.59 | 109.41 | 108.94 | 99.24 | 97.06 | 89.29 | 78.71 |   |   |   |
| Co-KO 12 | 100 | 102.9 | 103.28 | 101.42 | 96.17 | 89.62 | 83.17 |   |   |   |   |
|   |   |   |   |   |   |   |   |   |   |   |   |
|   |   |   |   |   |   |   |   |   |   |   |   |
| Experiment 3 |   |   |   |   |   |   |   |   |   |   |   |
| SFTSV-Ab (co.WT) 1 |
100 | 102.82 | 103.9 | 100.66 | 96.82 | 100.72 | 103.06 |   |   |   |   |
| SFTSV-Ab (co.WT) 2 |
100 | 98.94 | 87.13 | 86.88 | 78.81 | 77.5 | 76.13 | 71.56 | 73.63 | 71.44 | 72.38 |
| SFTSV-Ab (co.WT) 3 |
100 | 95.33 | 85.09 | 82.01 | 70.63 | 70.69 | 73.99 | 75.58 | 81.16 | 86.97 | 87.82 |
| SFTSV-Ab (co.WT) 4 |
100 | 89.93 | 82.01 | 80.77 | 71.72 | 72.74 | 72.96 | 72.85 | 78.79 | 80.88 | 83.09 |
| SFTSV-Ab (co.WT) 5 |
100 | 99.52 | 90.31 | 88.94 | 79.49 | 78.24 | 80.86 | 79.07 | 88.47 | 90.84 | 91.56 |
| SFTSV-Ab (co.WT) 6 |
100 | 95.55 | 84.94 | 84.51 | 79.09 | 78.17 | 80.98 | 79.88 | 82.8 | 84.57 | 88.9 |
| SFTSV-Ab (co.WT) 7 |
100 | 101.82 | 88.5 | 85.04 | 81.16 | 76.88 |   |   |   |   |   |
| SFTSV-Ab (co.WT) 8 |
100 | 88.98 |   |   |   |   |   |   |   |   |   |
| SFTSV-Ab (co.WT) 9 |
100 | 94.4 | 85.74 | 82.57 | 78.55 | 72.44 |   |   |   |   |   |
| SFTSV-Ab (co.WT) 10 |
100 | 101.85 | 90.36 | 86.19 | 79.58 | 73.75 |   |   |   |   |   |
| SFTSV-Ab (co.WT) 11 |
100 | 96.7 | 89.85 | 87.78 | 89.18 | 85.09 |   |   |   |   |   |
| SFTSV-Ab (co.WT) 12 |
100 | 97.87 | 90.8 | 83.24 | 82.75 | 79.71 |   |   |   |   |   |
|   |   |   |   |   |   |   |   |   |   |   |   |
| SFTSV-Ab (co.Ab) 1 |
100 | 99.94 | 90.78 | 88.37 | 79.85 | 75.14 |   |   |   |   |   |
| SFTSV-Ab (co.Ab) 2 |
100 | 97.2 | 87 | 85.66 | 76.63 | 77.39 |   |   |   |   |   |
| SFTSV-Ab (co.Ab) 3 |
100 | 96.17 | 86.13 | 81.29 | 74.72 | 70.53 |   |   |   |   |   |
| SFTSV-Ab (co.Ab) 4 |
100 | 100.35 | 92.35 | 91.65 | 81.32 | 77.35 |   |   |   |   |   |
| SFTSV-Ab (co.Ab) 5 |
100 | 94.09 |   |   |   |   |   |   |   |   |   |
| SFTSV-Ab (co.Ab) 6 |
100 | 100.38 | 90.87 | 87.48 | 83.57 | 80.18 |   |   |   |   |   |
| SFTSV-Ab (co.Ab) 7 |
100 | 101.54 | 88.76 | 86.92 | 79.47 | 77.87 | 79.17 | 81.12 | 89.82 | 96.45 | 97.87 |
| SFTSV-Ab (co.Ab) 8 |
100 | 96.57 | 84.36 | 83.81 | 75.69 | 79.42 | 82.01 | 80.14 | 86.1 | 87.85 | 95.19 |
| SFTSV-Ab (co.Ab) 9 |
100 | 99.4 | 101.68 | 101.2 | 96.71 | 100.9 | 100.12 | 94.85 | 96.41 | 98.2 | 98.2 |
| SFTSV-Ab (co.Ab) 10 |
100 | 90.12 | 78.35 |   |   |   |   |   |   |   |   |
| SFTSV-Ab (co.Ab) 11 |
100 | 98.21 | 90.66 | 87.53 | 78.64 | 75.45 | 77.24 | 75.9 | 82.16 | 85.29 | 90.79 |
| SFTSV-Ab (co.Ab) 12 |
100 | 96.87 | 88.4 | 87.21 | 83.32 | 86.14 | 86.39 | 87.08 | 91.29 | 96.99 | 96.55 |
|   |   |   |   |   |   |   |   |   |   |   |   |
| SFTSV-Ab (co.KO) 1 |
100 | 95.7 | 85.91 | 83.23 | 73.75 | 80.05 | 78.8 | 79.55 | 82.61 | 82.17 | 81.05 |
| SFTSV-Ab (co.KO) 2 |
100 | 98.18 | 91.92 | 87.66 | 81.84 | 81.9 | 81.15 | 83.78 | 87.35 | 88.92 | 91.42 |
| SFTSV-Ab (co.KO) 3 |
100 |   |   |   |   |   |   |   |   |   |   |
| SFTSV-Ab (co.KO) 4 |
100 | 96.03 | 86.17 | 83.08 | 73.84 | 71.48 | 71.85 | 73.09 | 76.44 | 78.43 | 80.6 |
| SFTSV-Ab (co.KO) 5 |
100 | 96.6 | 88.97 | 85.76 | 78.64 | 76.97 | 76.2 |   |   |   |   |
| SFTSV-Ab (co.KO) 6 |
100 | 112.36 |   |   |   |   |   |   |   |   |   |
| SFTSV-Ab (co.KO) 7 |
100 | 86.62 | 77.21 | 72.47 |   |   |   |   |   |   |   |
| SFTSV-Ab (co.KO) 8 |
100 | 95.62 | 85.78 | 83.42 | 77.26 | 76.22 |   |   |   |   |   |
| SFTSV-Ab (co.KO) 9 |
100 | 97.49 | 86.98 | 84.72 | 80.68 | 81.72 |   |   |   |   |   |
| SFTSV-Ab (co.KO) 10 |
100 | 91.28 | 80.08 | 74.48 | 68.24 |   |   |   |   |   |   |
| SFTSV-Ab (co.KO) 11 |
100 | 100.12 | 90.34 | 87.8 | 77.05 | 75.6 |   |   |   |   |   |
| SFTSV-Ab (co.KO) 12 |
100 | 101.18 | 92.53 | 89.54 | 81.44 | 84.18 |   |   |   |   |   |
|   |   |   |   |   |   |   |   |   |   |   |   |
| Co-WT 1 | 100 | 100 | 102.46 | 103.57 | 101.64 | 103.81 | 100 | 100.64 | 101.23 | 104.22 | 98.95 |
| Co-WT 2 | 100 | 101.6 | 99.82 | 101.66 | 101.83 | 104.44 | 104.61 | 101.42 | 103.96 | 106.45 | 104.67 |
| Co-WT 3 | 100 | 102.78 | 100.95 | 102.84 | 102.21 | 101.58 | 101.58 | 96.53 | 99.12 | 101.07 | 98.74 |
| Co-WT 4 | 100 | 101.26 | 102.82 | 102.7 | 103.66 | 105.28 | 104.8 | 100.06 | 102.64 | 106.71 | 103.3 |
| Co-WT 5 | 100 | 102.36 | 100.91 | 101.99 | 99.21 | 106.71 | 106.28 | 101.93 | 103.99 | 111.72 | 106.65 |
| Co-WT 6 | 100 | 100.42 | 99.64 | 102.7 | 100.24 | 101.44 | 102.1 | 96.82 | 98.62 | 101.62 | 99.1 |
| Co-WT 7 | 100 | 99.35 | 99.12 | 97.24 | 97.83 | 97.36 |   |   |   |   |   |
| Co-WT 8 | 100 | 92.13 | 89.87 | 89.3 | 89.7 | 89.7 |   |   |   |   |   |
| Co-WT 9 | 100 | 109.71 | 104.92 | 103.16 | 104.92 | 101.58 |   |   |   |   |   |
| Co-WT 10 | 100 | 91.61 | 84.67 | 97.26 | 100.64 | 98.37 |   |   |   |   |   |
| Co-WT 11 | 100 | 89.95 | 94.55 | 86.83 | 86.77 | 90.07 |   |   |   |   |   |
| Co-WT 12 | 100 | 89.69 | 89.07 | 91.93 | 97.14 | 96.47 |   |   |   |   |   |
|   |   |   |   |   |   |   |   |   |   |   |   |
| Co-Ab 1 | 100 | 88.9 | 87.42 | 88.78 | 88.66 | 86.62 |   |   |   |   |   |
| Co-Ab 2 | 100 | 105.72 | 102.01 | 101.4 | 101.46 | 99.88 |   |   |   |   |   |
| Co-Ab 3 | 100 | 101.71 | 102.5 | 101.83 | 102.32 | 103.23 |   |   |   |   |   |
| Co-Ab 4 | 100 | 103.51 | 102.32 | 102.2 | 100.12 | 101.9 |   |   |   |   |   |
| Co-Ab 5 | 100 | 87.62 | 100.06 | 98.78 | 97.5 | 100.87 |   |   |   |   |   |
| Co-Ab 6 | 100 | 105.39 | 104.84 | 103.74 | 103.86 | 104.17 |   |   |   |   |   |
| Co-Ab 7 | 100 | 108.96 | 105.07 | 101.85 | 101.3 | 104.82 | 102.47 | 99.38 | 97.78 | 98.45 | 98.27 |
| Co-Ab 8 | 100 | 92.59 | 93.63 | 79.52 | 94.99 | 101.2 | 99.56 | 92.92 | 97.11 | 95.97 | 95.81 |
| Co-Ab 9 | 100 | 96.98 | 99.81 | 100.43 | 97.16 | 100.8 | 107.28 | 98.89 | 105.8 | 103.76 | 100.74 |
| Co-Ab 10 | 100 | 100.68 | 100 | 98.76 | 97.85 | 101.86 | 99.1 | 96.05 | 96.78 | 97.29 | 100 |
| Co-Ab 11 | 100 | 95.14 |   |   |   |   |   |   |   |   |   |
| Co-Ab 12 | 100 | 104.89 | 102.11 | 104.47 | 103.69 | 106.22 | 105.26 | 103.63 | 104.23 | 104.53 | 102.72 |
|   |   |   |   |   |   |   |   |   |   |   |   |
| Co-KO 1 | 100 | 94.9 | 95.36 | 109.02 | 96.14 | 100.52 | 100.85 | 100.26 | 103.33 | 106.67 | 103.92 |
| Co-KO 2 | 100 | 101.91 | 101.25 | 88.55 | 83.74 | 85.71 | 89.4 | 91.18 | 96.38 | 94.67 | 98.75 |
| Co-KO 3 | 100 | 101.97 | 96.26 | 100.06 | 98.03 | 96.83 | 95.12 | 96.64 | 101.65 | 105.45 | 103.36 |
| Co-KO 4 | 100 | 94.54 |   |   |   |   |   |   |   |   |   |
| Co-KO 5 | 100 | 102.02 | 99.49 | 99.27 | 99.61 | 99.66 | 98.26 | 94.73 | 103.09 | 105.11 | 102.13 |
| Co-KO 6 | 100 | 98 | 98.38 | 101.94 | 101.44 | 101.38 | 100.63 | 97.75 | 101.13 | 102.75 | 99.38 |
| Co-KO 7 | 100 | 101.01 | 97.09 | 97.47 | 92.15 | 91.96 |   |   |   |   |   |
| Co-KO 8 | 100 | 104.2 | 99.63 | 99.38 | 93.14 | 95.43 |   |   |   |   |   |
| Co-KO 9 | 100 | 100.78 | 96.63 | 98.32 | 95.28 | 99.61 |   |   |   |   |   |
| Co-KO 10 | 100 | 102.63 | 100.81 | 102.81 | 97.63 | 97.31 |   |   |   |   |   |
| Co-KO 11 | 100 | 109.14 | 99.59 | 100.88 | 87.26 | 84.54 |   |   |   |   |   |
| Co-KO 12 | 100 | 102.21 | 99.19 | 97.09 | 94.82 | 100.35 |   |   |   |   |   |

Reviewer 2 Report
Comments and Suggestions for Authors
-Authors described that IFNAR Ab and IFNAR KO as spreaders exhibited higher transmissibility to co-housed mice than WT. Moreover, IFNAR KO as recipients were more susceptible to SFTSV infection than WT in line 23 and 25.
: Could authors more explain that IFNAR KO mice are more susceptible to SFTSV infection than IFNAR Ab?
- Authors described that Aging is associated with multiple defects to produce interferon in response to viral infection [11] in line 42 and 43
: Age is one of the important factors.
Could authors comment age among IFNAR KO mice, IFNAR Ab, and WT in material and methods?
- Authors described below in this manuscript.
In line 304 and 305, the present study revealed that immunocompromised mice without IFNAR signaling showed higher susceptibility to SFTSV than normal mice.
In line 365 and 366, Type I interferon signaling may be a pivotal factor for the transmissibility of SFTSV without a 366 vector such as a tick.
: Could authors show (or describe) the role (or mechanism) of IFNAR signaling in SFTSV infection and put reference(s)?
- Authors described that In SFTSV transmission experiments, IFNAR Ab mice might have performed better as spreaders than WT and IFNAR KO mice. Despite having a lower mortality rate and higher SFTS viral loads than WT, IFNAR Ab mice offered an advantage of allowing experiments to be designed for longer durations in line 358 and 361.
: Could authors describe the reason (why IFNAR Ab (anti-type I interferon-α receptor blocking antibody) mice are better spreaders than WT and IFNAR KO mice) in this part ?
Comments on the Quality of English LanguageAcceptable
Author Response
<<Cover letter>>
We would like to thank Editors and Reviewers for careful evaluation on our manuscript. We agree the reviewers’ comments are insightful and their suggestions are excellent. We have revised the manuscript extensively as suggested by the reviewers. We hope this revised manuscript is suitable for publication in viruses. Thank you for your consideration of our manuscript. We look forward to hearing your and reviewers’ comments. The following is our point by point response to the reviewer’s comments.
<<Point by point response>>
1.-Authors described that IFNAR Ab and IFNAR KO as spreaders exhibited higher transmissibility to co-housed mice than WT. Moreover, IFNAR KO as recipients were more susceptible to SFTSV infection than WT in line 23 and 25.
Thank you pointing this out. As indicated in lines 246-248, a body weight loss was observed only in the IFNAR KO group. As shown in Figure 4, at 9dpi, the viral load in the serum of IFNAR KO was higher compared to the other co-housed groups and the average value for IFNAR KO was similar to that of the SFTSV-Ab group. In histopathologically, only IFNAR KO group observed white pulp depletion in spleen at 5 dpi and severe inflammatory cell infiltration and foci of necrotic lesions in liver. These results suggested that IFNAR KO mice are more susceptible to SFTSV infection than IFNAR Ab mice.
2.- Authors described that Aging is associated with multiple defects to produce interferon in response to viral infection [11] in line 42 and 43
: Age is one of the important factors.
Could authors comment age among IFNAR KO mice, IFNAR Ab, and WT in material and methods?
We thank the reviewer for bringing up this point. All mice were 5 weeks old and weighed similarly, weighing 15 to 16 g. As suggested by reviewer’s comments, we added ’five-week-old’ in page 2, line 94.
3.- Authors described below in this manuscript.
In line 304 and 305, the present study revealed that immunocompromised mice without IFNAR signaling showed higher susceptibility to SFTSV than normal mice.
In line 365 and 366, Type I interferon signaling may be a pivotal factor for the transmissibility of SFTSV without a 366 vector such as a tick.
: Could authors show (or describe) the role (or mechanism) of IFNAR signaling in SFTSV infection and put reference(s)?
Thank you for pointing this out. Previous research data showed that SFTSV has the capability to disrupt the initial induction of type I IFNs by targeting host kinases TBK1/IKKε [1-3]. Additionally, SFTSV has been observed to inhibit type I IFN-triggered signaling pathways and interferon stimulated gene expression [4]. The contents were added to the discussion (Page 10, line 322-327).
4.- Authors described that In SFTSV transmission experiments, IFNAR Ab mice might have performed better as spreaders than WT and IFNAR KO mice. Despite having a lower mortality rate and higher SFTS viral loads than WT, IFNAR Ab mice offered an advantage of allowing experiments to be designed for longer durations in line 358 and 361.
: Could authors describe the reason (why IFNAR Ab (anti-type I interferon-α receptor blocking antibody) mice are better spreaders than WT and IFNAR KO mice) in this part ?
We thank the reviewer for bringing up this point. WT not only has very low pathogenicity to progress to the role of a spreader, but also has a low transmissibility. According the results of the first experiment (Figure 2), even though WT were cohoused with the most SFTSV-susceptible IFNAR KO mice, there was no death or clinical symptoms in the IFNAR KO group. KO died within 4 dpi even though it was infected with a low titers of SFTSV [5]. Based on these results, it is difficult to observe virus shedding and transmission in IFNAR KO, so it is not suitable for the role of a spreader. On the other hand, the IFNAR Ab group has a lower mortality rate than the IFNAR KO group, but shows higher pathogenicity than WT. Therefore, using the IFNAR Ab group as a spreader, longer-term experiments can be conducted and higher virus transmission can be observed.
<Reference>
- Ning, Y.-J., et al., Viral suppression of innate immunity via spatial isolation of TBK1/IKKε from mitochondrial antiviral platform. Journal of molecular cell biology, 2014. 6(4): p. 324-337.
- Wu, X., et al., Evasion of antiviral immunity through sequestering of TBK1/IKKε/IRF3 into viral inclusion bodies. Journal of virology, 2014. 88(6): p. 3067-3076.
- Santiago, F.W., et al., Hijacking of RIG-I signaling proteins into virus-induced cytoplasmic structures correlates with the inhibition of type I interferon responses. Journal of virology, 2014. 88(8): p. 4572-4585.
- Ning, Y.-J., et al., Disruption of type I interferon signaling by the nonstructural protein of severe fever with thrombocytopenia syndrome virus via the hijacking of STAT2 and STAT1 into inclusion bodies. Journal of virology, 2015. 89(8): p. 4227-4236.
- Park, S.C., et al., Pathogenicity of severe fever with thrombocytopenia syndrome virus in mice regulated in type I interferon signaling: Severe fever with thrombocytopenia and type I interferon. Lab Anim Res, 2020. 36: p. 38.

Round 2
Reviewer 2 Report
Comments and Suggestions for Authors
NOTHING
Comments on the Quality of English LanguageNOTHING